# Efficacy of Large Use of Combined Hypofractionated Radiotherapy in a Cohort of Anti-PD-1 Monotherapy-Treated Melanoma Patients

**DOI:** 10.3390/cancers14174069

**Published:** 2022-08-23

**Authors:** Philippe Saiag, Rafaele Molinier, Anissa Roger, Blandine Boru, Yves Otmezguine, Joelle Otz, Charles-Ambroise Valery, Astrid Blom, Christine Longvert, Alain Beauchet, Elisa Funck-Brentano

**Affiliations:** 1Department of General and Oncologic Dermatology, Ambroise Paré Hospital, APHP, & EA 4340 “Biomarkers in Cancerology and Hemato-Oncology”, UVSQ, Université Paris-Saclay, 92104 Boulogne-Billancourt, France; 2Department of Radiology, Ambroise Paré Hospital, APHP, 92104 Boulogne-Billancourt, France; 3Oncology Centre, Porte de Saint-Cloud Clinic, 92100 Boulogne-Billancourt, France; 4Department of Radiotherapy, Curie Hospital, 92210 Saint-Cloud, France; 5Department of Neuroradiotherapy, Pitié-Salpêtrière Hospital, APHP, 75013 Paris, France; 6Department of Public Health, Ambroise Paré Hospital, APHP & UVSQ, Université Paris-Saclay, 92104 Boulogne-Billancourt, France

**Keywords:** melanoma, anti-PD-1 antibody, nivolumab, pembrolizumab, radiotherapy, abscopal effect

## Abstract

**Simple Summary:**

Melanoma patients who failed anti-PD-1 therapy have limited therapeutic options. Some studies suggested the efficacy of radiotherapy combined with anti-PD-1 monoclonal antibodies. We previously reported in small-sized studies that hypofractionated radiotherapy combined with an unmodified anti-PD-1 monotherapy regimen induced long-lasting efficacy. This study shows that the large use of hypofractionated radiotherapy combined with anti-PD-1 induced high rates of complete response (32.5% [95% CI: 26.1–38.9]) in a cohort of 206 melanoma patients. Radiated patients who had confirmed failure to anti-PD-1 monotherapy had longer progression-free and overall survival than non-radiated ones. No unexpected safety concern was observed. Although no clinical predictive factors have been identified in our study, a synergy between anti-PD1 and radiotherapy is likely. Adding hypofractionated radiotherapy in metastatic melanoma patients treated with anti-PD-1 is a safe option, which may increase the response rate and could be offered in patients with lesions threatening or not responding to anti-PD1.

**Abstract:**

To assess the role of radiotherapy in anti-PD-1-treated melanoma patients, we studied retrospectively a cohort of 206 consecutive anti-PD-1 monotherapy-treated advanced melanoma patients (59% M1c/d, 50% ≥ 3 metastasis sites, 33% ECOG PS ≥ 1, 33% > 1st line, 32% elevated serum LDH) having widely (49%) received concurrent radiotherapy, with RECIST 1.1 evaluation of radiated and non-radiated lesions. Overall (OS) and progression-free (PFS) survivals were calculated using Kaplan–Meier. Radiotherapy was performed early (39 patients) or after 3 months (61 patients with confirmed anti-PD-1 failure). The first radiotherapy was hypofractionated extracranial radiotherapy to 1–2 targets (26 Gy-4 weekly sessions, 68 patients), intracranial radiosurgery (25 patients), or palliative. Globally, 67 (32.5% [95% CI: 26.1–38.9]) patients achieved complete response (CR), with 25 CR patients having been radiated. In patients failing anti-PD-1, PFS and OS from anti-PD-1 initiation were 16.8 [13.4–26.6] and 37.0 months [24.6–NA], respectively, in radiated patients, and 2.2 [1.5–2.6] and 4.3 months [2.6–7.1], respectively, in non-radiated patients (*p* < 0.001). Abscopal response was observed in 31.5% of evaluable patients who radiated late. No factors associated with response in radiated patients were found. No unusual adverse event was seen. High-dose radiotherapy may enhance CR rate above the 6–25% reported in anti-PD-1 monotherapy or ipilimumab + nivolumab combo studies in melanoma patients.

## 1. Introduction

Immune checkpoint inhibitors (ICI) have improved advanced melanoma patients’ treatment [1], particularly the anti-programmed death-1 (PD-1) monoclonal antibodies (mAb) nivolumab and pembrolizumab, which induced, in the first-line setting, longer progression-free survival (PFS) and overall survival (OS) than chemotherapy or ipilimumab, an anti-CTLA-4 mAb [2,3]. However, depending on treatment line [4] and length of follow-up, anti-PD-1 monotherapy was associated with a complete (CR) + partial (PR) response rate of only 27–52% and a median PFS of 3.1–6.9 months [2,3,4,5,6] in patients without active brain metastasis. Moreover, intracranial metastases, reported in 40–50% of patients, convey a poor prognosis [7]. Thus, improved strategies are required. Formal demonstration of improved PFS and OS by nivolumab and ipilimumab combination, when compared to nivolumab alone, is lacking and severe adverse events (AEs) are very frequent [6]. This combination also provided encouraging PFS and OS data and high response rates in melanoma patients with 1–4 asymptomatic intracranial metastases [8,9].

Radiotherapy (RT) exerts multiple vascular, stromal, and immunological changes in the tumor microenvironment [10]. Preclinical data suggest that RT combined with ICI may improve tumor response [11]. Some melanoma studies suggested the efficacy of RT combined with anti-PD-1 mAb, but its place in melanoma care is debated because of a limited number of prospective trials [12,13], uncontrolled design, and the use of either multisite RT [14], which precluded the evaluation of the abscopal effect, or suboptimal radiation schedules (RT delivered before anti-PD-1 initiation, insufficient dosing/session) [15,16]. Moreover, RT was frequently combined with anti-PD-1 mAb early on [16,17,18], and responses could therefore originate solely from the late efficacy of anti-PD-1.

Hypofractionated RT delivers higher doses per session than standard palliative RT and requires a reduced number of sessions. We previously reported that hypofractionated RT combined with an unmodified anti-PD-1 mAb monotherapy regimen induced long-lasting efficacy, with a CR+PR rate of 36–38% in melanoma patients who either received this combination early [19] because of life-threatening metastases or had previously failed anti-PD-1 monotherapy [20].

Herein, our objectives were to further investigate this combination in real-life conditions, including in patients with active brain metastases. Could widespread use of extracranial hypofractionated RT and/or intracranial stereotactic radiosurgery (SRS), which also delivers high doses per session, combined with anti-PD-1 monotherapy in a vast cohort of consecutive anti-PD-1 monotherapy-treated melanoma patients, enhance the CR rate above the 6–25% reported in melanoma registration trials [2,3,4,5,6] (with mainly first-line patients without active brain metastasis)? This is an important goal, as patients experiencing CR have a low recurrence probability after anti-PD-1 discontinuation [21]. Secondary objectives were (1) to compare PFS and OS in radiated and non-radiated patients in the whole population and those failing anti-PD-1 mAb; (2) to characterize the profile of radiated patients achieving CR+PR; (3) to assess safety.

## 2. Materials and Methods

This was a monocenter retrospective analysis of data prospectively collected according to previously published procedures [19,20] in our referral skin cancer department for anti-PD-1 mAb-treated melanoma patients not included in double-blinded trials. Nivolumab or pembrolizumab was infused intravenously according to product labels until unambiguous progressive disease (PD), unacceptable AE, or decision to discontinue treatment, but was continued beyond progression at first evaluation to allow for pseudo-progression [22], or later if ≥1 lesion could be treated locally. AEs were graded according to the Common Terminology Criteria for Adverse Events (CTCAE) version 4.0. We evaluated efficacy at least every 3 months using thoracic, abdominal, and pelvic computed tomography (CT) scans, head CT scans, or magnetic resonance imaging, carried out by melanoma-experienced radiologists, during treatment and 5 years after anti-PD-1 discontinuation. In addition, normal ^18^F-labeled fluorodeoxyglucose positron emission tomography (FDG-PET) scans were required to confirm CR or to address ambiguous CT images.

Images were reviewed during a weekly joint meeting with radiologists through tumor evaluations according to RECIST 1.1 [23]. CR was defined as the disappearance of all lesions (with the smallest axis of lymph nodes <10 mm), PR as a decrease by >30% of the sum of the diameters of the target lesions, PD as an increase >20% of this sum or occurrence of any new lesions, and stable disease (SD) as no sufficient shrinkage to qualify for PR nor sufficient increase to qualify for PD. Patients who died before the first evaluation were also qualified as PD. The abscopal effect was defined as PR or CR outside radiated fields.

Our tumor board provided indications for RT, which was performed in combination with an unchanged anti-PD-1 regimen either within the first 3 months of PD-1 blockade for rapidly progressing, symptomatic, or life-threatening lesions, or later in patients with confirmed PD on two consecutive computed tomography (CT)-scans (to rule out pseudo-progression) or with long-lasting SD. RT was standardized, with 20–26 Gy delivered in 3–5 weekly sessions for extracranial lesions and SRS in one or two sessions, delivered through a Gamma-knife, for intracranial lesions.

For this study, we analysed records of all patients with confirmed advanced melanoma who initiated anti-PD-1 monotherapy, regardless of BRAF/NRAS mutational status and previous therapies, between 1 January 2014 and 30 August 2019. Key exclusion criteria were age < 18 years and association with ipilimumab. The database was locked on 1 August 2021, and we converted all recorded American Joint Committee on Cancer (AJCC) stages to the 8th edition [24]. The primary endpoint was the CR rate. The secondary endpoints were the response rates of radiated and non-radiated lesions (to estimate the abscopal effect), as well as PFS (time from the first dose to confirmed PD after anti-PD-1 monotherapy or RT for radiated patients, or death), OS (time from the first infusion of anti-PD1 mAb to death), and safety. This study updates and expands our previous works, which included only 25 [19] and 26 patients [20].

STROBE guidelines for observational studies were followed. The Kaplan–Meier method was used to estimate survival rates. Logistic regressions were used to study relationships between CR+PR after RT and the following parameters: BRAF mutational status, LDH serum levels, number of metastatic sites, treatment-naïve status or not, Eastern Cooperative Oncology Group (ECOG) performance status, extracranial or intracranial RT, AJCC staging, oligometastatic disease (defined as ≤5 metastases) at anti-PD-1 initiation, and oligoprogression (defined as <5 metastases at progression) at first RT. A *p* value < 0.05 was considered statistically significant. Statistical analysis was carried out using SPSS v. 24 (IBM Corp, Armonk, NY, USA) and R v. 3.4.3 (R Core Team, Vienna, Austria: R Foundation for Statistical Computing, 2017). The sample size calculation showed that with >200 patients treated, of whom ≥100 received concurrent radiotherapy, we had enough power to demonstrate an increase of the CR rate from 15–20% to 30%.

According to French law, this study abided by standard medical practices and did not require written informed consent nor a formal approval by a national ethics committee. Nonetheless, verbal consent was obtained from all living patients, and the protocol (Appendix A) was accepted by the research ethics committee of Paris-Saclay University (CER, number 257). Statistical analysis was conducted following this protocol, which was established before data collection. The study was conducted according to the principles of the Declaration of Helsinki [25]. The anonymized datasets of this study are available from the corresponding author upon request.

## 3. Results

During the study time frame, 206 consecutive patients initiated nivolumab (83%) or pembrolizumab (17%) monotherapy. Median duration on anti-PD-1 was 8 months (range: 1–91). Most patients had severe disease: 59% M1c+d disease, 50% ≥3 metastatic sites, 33% ECOG PS ≥1, 33% non-treatment-naïve, 32% LDH >upper limit of normal, 32% liver or 22% intracranial metastases. Median follow-up was 22 months (range: 1–79), without any lost to follow-up.

One hundred patients (49%) were very severe and received concurrent RT, either early (<3 months of PD-1 blockade for rapidly progressing, symptomatic, or life-threatening lesions (*n* = 39)) or late (>3 months for confirmed PD (*n* = 50) or more occasionally for long-lasting SD (*n* = 3)), or both early and late (*n* = 8). Table 1 shows patients’ characteristics at anti-PD-1 initiation.

Radiated and non-radiated patients lacked statistically significant baseline differences except for liver metastases and AJCC staging. Notably, only 20% of radiated patients had oligometastatic disease and 21% had oligoprogression. The first combined RT was hypofractionated RT targeting 1–2 extracranial lesions (*n* = 68), SRS for intracranial metastases (*n* = 25), or standard palliative RT (Table 2). A total of 39 patients later received a second RT for persistent PD, targeting a previously non-radiated target lesion at a median of 7.2 months [range: 2–17] after the first one (Table 2).

Overall, 67 out of 206 patients (32.5% [95% CI: 26.1–38.9]) achieved CR (including seven with surgical resection of a residual lesion), and anti-PD-1 was discontinued in 64 patients with CR, of whom 14 (22%) had relapsed after a median of 37 months [range: 8–75] off anti-PD-1. RT had been performed in 25 patients achieving CR (10 and 15 with early or late RT, respectively), while 42 patients with CR were not radiated. We also observed 34 PR (17%), 15 SD (7%), and 90 PD (44%).

Table 3 shows the response rates in radiated and non-radiated areas for radiated patients. Of note, eight (33%) of the radiated patients achieving CR and six (55%) of those with PR required two sessions of RT. Among the 61 patients with late RT (because of anti-PD-1 failure), 54 had evaluable lesions outside radiated areas: abscopal response was observed in non-radiated lesions in 17 patients (31.5%).

For the whole cohort, median PFS and OS were 16.3 [95% CI: 9.4–NA] and 23.7 [16.8–35.9] months, respectively, without significant difference in non-radiated and radiated patients (Figure 1). When restricting analysis to the 61 patients radiated for anti-PD-1 failure and to the 51 non-radiated patients also with anti-PD-1 failure, median PFS and OS were 16.8 [13.4–26.6] and 37.0 [24.6–NA] months, respectively, versus 2.2 [1.5–2.6] and 4.3 months [2.6–7.1], respectively (*p* < 0.001) (Figure 2). We also studied the role of hypofractionated extracranial radiotherapy: when restricting the analysis to the 75 patients who received hypofractionated radiotherapy and to the 106 non-radiated patients, median PFS and OS were 16.2 [12.2–26.6] and 31.0 [23.7–41.9] months, respectively, in radiated patients, versus not achieved and 20.4 [11.2–52.7] months, respectively (*p* = 0.054 for OS, 0.16 for PFS) (Figure 3). Overall, PFS and OS curves seemed to have reached a plateau in radiated patients. Appendix A shows PFS and OS curves from date of RT in radiated patients.

In radiated patients, the following parameters collected at anti-PD-1 initiation were not associated with achieving CR+PR: BRAF mutational status, AJCC staging, naïve versus non-naïve status, ECOG performance status (0 versus ≥1), LDH serum level (normal versus >normal), presence of <3 metastatic sites versus ≥3 or of oligometastatic disease. Use of extracranial RT versus SRS, early versus late RT, and oligoprogression were also not associated with CR+PR.

A total of 29.1%, 9.7%, 1%, and 0.5% of patients experienced grade 1–2, 3, 4, or 5 AEs, respectively, without unusual ones. RT-related AEs included 5% asthenia or digestive symptoms, 4% grade 2 dermatitis, 1% RT-induced pneumonitis, 1% radiation-induced brain necrosis.

## 4. Discussion

We report herein a retrospective analysis of data collected prospectively in a cohort of 206 consecutive anti-PD-1 monotherapy-treated melanoma patients with 49% of patients receiving hypofractionated RT combined with an unmodified anti-PD-1 regimen. We observed a 32.5% [95% CI: 26.1–38.9] CR rate, which is above figures obtained in melanoma patients receiving anti-PD-1 monotherapy [2,3,4,5,6] or combined ipilimumab + nivolumab [26], despite numerous patients with active intracranial metastasis and/or non-treatment-naïve status being included. The high percentage of AJCC M1c and M1d patients, with high LDH serum levels, ECOG performance status >1, or liver metastasis, which all represent poor prognostic factors [27], also highlights real-life patients’ severity. Benefits of RT were observed particularly in patients failing anti-PD-1 blockade. Anti-PD-1 treatment could be withdrawn in 97% of patients with CR. Thus, combining anti-PD-1 with RT could represent an opportunity for patients failing anti-PD-1 blockade.

Limitations of our study include its retrospective nature, the absence of an independent evaluation of images, and the absence of formal comparison with patients treated beyond PD with anti-PD-1 mAb alone [28]. Moreover, evaluations retained for radiated patients were those performed after radiation, which induced a systematic bias. Thus, the potential benefits elicited by this combination are not formally established. However, to the best of our knowledge, this represents the largest series of combined RT and anti-PD-1 blockade, and we are confident with the CR we observed, as our recurrence rate of 22% after a median of 37 months off therapy compares favorably with the 14% observed after a median 18 months of follow-up in patients achieving CR [21].

Our results obtained in patients radiated late because of anti-PD-1 failure are likely not due to late efficacy of ICI and suggest a true synergy of the combination, as (1) we included patients after the exclusion of pseudo-progression [22]; (2) the median delay between anti-PD-1 mAb initiation and the first day of RT was 6.1 months in these patients, whereas most anti-PD-1-treated patients who later achieve CR demonstrate PR at the 3-month evaluation [29]; (3) an abscopal effect was observed in 31.5% of these patients, a feature which was infrequently reported before the ICI era [30]; and (4) there are robust preclinical data (cf. infra).

The absence of a statistically significant difference in PFS and OS between radiated and non-radiated patients in the whole population despite similar characteristics at inclusion for most prognostic criteria (Table 1) can be explained by our indications of RT, which required initial life-threatening or symptomatic metastases (39% of radiated patients), later anti-PD-1 initial monotherapy failure (53%), or a combination of both situations (8%), leading to a poorer prognosis in the radiated population. This reinforces the value of our findings.

Previous [12,13,14,15,16,17,18,19,20] and present results favor the hypothesis that hypofractionated RT delivering fractions of ≥6 Gy represents a suitable extracranial RT regimen in anti-PD-1-treated melanoma patients. In cell culture and animal models, radiation-induced immunogenic cell death occurred through the release of damage-associated molecular patterns, such as double-stranded DNA, with the secretion of type-1 interferons (IFN) through the GMP–AMP synthase/stimulator of IFN genes (cGAS/STING) pathway [11]. DNA exonuclease 3′ repair exonuclease 1 (Trex1), induced by radiation doses > 12 Gy, degraded double-stranded DNA, abolished type-1 IFN release, and finally radiation immunogenicity [31]. Fractionated RT was more effective in inducing immune-mediated effects than a single ablative dose and may overcome RT-induced adaptive resistance [32]. Repeated radiations at doses (8 Gyx3) above the threshold of Trex1 induction greatly amplified type-1 IFN production, resulting in recruitment and activation of Batf3-dependent dendritic cells, which are essential for priming of CD8+ T cells that induce tumor rejection in the context of ICI [31]. Notably, when restricting the analysis to the patients who received hypofractionated extracranial radiotherapy and to the non-radiated patients, OS tended to be longer in radiated patients versus non-radiated ones, with a *p*-value of 0.054. Optimal dosing and fractionation strategy for each cancer type has not yet been determined, but larger doses per fraction, as in our series, were associated with enhanced abscopal effects [10]. Moreover, a trial of ipilimumab combined with hypofractionated RT (6 Gyx5 or 9 Gyx3) in patients with non-small-cell lung cancer (for which ipilimumab had failed to demonstrate any efficacy) provided a proof-of-concept, with radiological responses observed along with type-1 IFN release and radiation-induced transcriptional upregulation of neo-antigens [33]. Solid data on dose-ranging for combined ICI and SRS in melanoma patients are lacking, but the tumor microenvironment is likely different in the brain. Finally, in addition to hypofractionated RT at a high fraction per session, repetition of RT while on anti-PD-1 mAb may also contribute to our findings, as repetition of RT on a different target occurred in 40% of our radiated patients achieving CR or PR. Treating wider tumor areas with RT combined with ICI has been shown to improve mucosal melanoma control [34].

This large series confirms the tolerability of combined RT and anti-PD-1 mAb [12,17,18,19,34], as no unusual new nor higher frequency of anti-PD-1-related or RT-induced AEs were observed.

We did not find significant clinical predictive factors for CR+PR during combination therapy but did not assay potential biomarkers relevant for investigating combined ICI and RT [11,32] as none of these have been validated.

Although we included consecutive patients during a long period in a real-life setting, the external validity of our results should be demonstrated in other settings using similar hypofractionated RT regimens. This approach should also be investigated in anti-PD-1+anti-CTLA-4 or anti-PD-1+anti-LAG3–3 [35] refractory patients.

## 5. Conclusions

We suggest a synergy between anti-PD1 and RT, which could be due to tumor neoantigens release after irradiation stimulating the anti-tumor immune response [33]. High-dose hypofractionated RT may enhance anti-PD-1 efficacy by enhancing the CR rate above 30% in melanoma patients, thus allowing safe anti-PD-1 cessation. Controlled studies are needed.

## Figures and Tables

**Figure 1 cancers-14-04069-f001:**
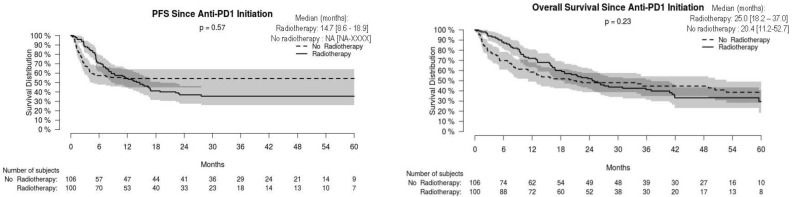
Progression-free (PFS) and overall (OS) survival in radiated and non-radiated anti-PD-1-treated melanoma patients.

**Figure 2 cancers-14-04069-f002:**
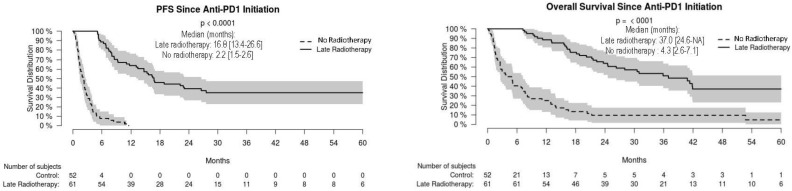
Progression-free (PFS) and overall (OS) survival in radiated and non-radiated melanoma patients with anti-PD-1 treatment failure.

**Figure 3 cancers-14-04069-f003:**
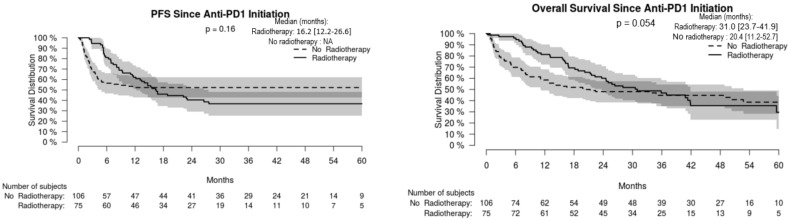
Progression-free (PFS) and overall (OS) survival in radiated melanoma patients having received hypofractionated radiotherapy and non-radiated melanoma patients.

**Table 1 cancers-14-04069-t001:** Characteristics at anti-PD-1 initiation for radiated and non-radiated patients.

	With Radiotherapy (*n* = 100)	Without Radiotherapy (*n* = 106)	*p*
Age, years (mean ± SD)	68.5 (±14.6)	69.3 (±15.2)	0.70
median (range)	69 (32–94)	71 (32–92)	
Male sex, *n* (%)	62 (62)	70 (66)	0.65
Primary melanoma location, *n* (%)			0.16
Head and neck	22 (22)	21 (20)	
Trunk	26 (16)	38 (36)	
Upper limb	8 (8)	14 (13)	
Lower limb	31 (31)	27 (25)	
No primary	13 (13)	6 (6)	
Breslow, mm (±SD)	4.9 (±4.7) (*n* = 76)	4.8 (±4.2) (*n* = 95)	0.92
Ulceration of primary melanoma, *n* (%)			0.09
Yes	39 (39)	53 (50)	
No	33 (33)	38 (36)	
No primary	13 (13)	6 (6)	
Unknown	15 (15)	9 (9)	
AJCC staging *, *n* (%)			0.02
III non-resectable	13 (13)	7 (7)	
IV M1a	15 (15)	12 (11)	
IV M1b	13 (13)	24 (23)	
IV M1c	31 (31)	46 (43)	
IV M1d	28 (28)	17 (16)	
Line of treatment, *n* (%)			0.67
1	68 (68)	70 (66)	
2	25 (25)	25 (24)	
3	6 (6)	7 (7)	
≥4	1 (1)	4 (4)	
Previous systemic therapy, *n* (%)			
BRAFinh + MEKinh	23 (23)	25 (24)	1
Ipilimumab	5 (5)	9 (9)	0.47
Chemotherapy	6 (6)	8 (8)	0.65
Other	2 (2)	3 (3)	1
Organs with metastases, *n* (%)			1
<3	50 (50)	53 (50)	
≥3	50 (50)	53 (50)	
Presence of liver metastasis, *n* (%)	23 (23)	43 (41)	0.01
ECOG performance status, *n* (%)			0.80
0	67 (67)	71 (67)	
1	27 (27)	26 (24)	
2	3 (3)	6 (6)	
3	1 (1)	2 (2)	
Unknown	2 (2)	1 (1)	
LDH serum level, *n* (%)			0.50
Normal	72 (72)	66 (62)	
>1 and <2 upper limit of normal	20 (20)	30 (28)	
>2 upper limit of normal	7 (7)	8 (8)	
Unknown	1 (1)	2 (2)	

Data are numbers (percentage); SD = standard deviation; AJCC = American Joint Committee on Cancer; ECOG = Eastern Cooperative Oncology Group Performance status; LDH = lactate dehydrogenase; BRAFinh + MEKinh: association of a BRAF-inhibitor and an MEK-inhibitor; * AJCC 8th edition.

**Table 2 cancers-14-04069-t002:** Characteristics of the radiotherapy series in the 100 radiated and anti-PD-1-treated patients.

	Patients, *n* (%)
**First series of radiotherapy**	100 (100)
Hypofractionated, extracranial	68 (68)
Median total dose Gy (range)	26 (6.5–27.5)
Median number of sessions (range)	4 (1–5)
Median interval between sessions, days	7
Radiated fields	
Soft tissues & lymph nodes	42 (42)
Chest or mediastinum	11 (11)
Bone	9 (9)
Retroperitoneum or intra-abdominal	6 (6)
Radiosurgery on brain lesions	25 (25)
Number of sessions (range)	1 (1–3)
Median number of brain lesions treated (range)	2 (1–10)
Other *	7 (7)
**Second series of radiotherapy**	39 (39)
Hypofractionated, extracranial	21 (21)
Median total dose Gy (range)	26 (10–26)
Median number of sessions (range)	4 (2–6)
Median interval between sessions, days	7
Radiosurgery on brain lesions	15 (15)
Standard palliative radiotherapy	3 (3)

Unless specified, data are numbers (percentage). * Others: standard palliative radiotherapy (30 Gy, 10 sessions, 5 patients), proton therapy or contact therapy (1 patient each).

**Table 3 cancers-14-04069-t003:** Best response after radiotherapy and anti-PD-1 monoclonal antibodies combination in melanoma patients who failed anti-PD-1 monotherapy or with rapidly progressing disease at anti-PD-1 initiation.

	Global Response (Total *n* = 100)	Radiated Area	Non-Radiated Area
CR ^a^	25 (25)	26 (26)	19 (19)
PR ^b^	10 (10)	26 (26)	13 (13)
SD	1(1)	8 (8)	0 (0)
PD	64 (64)	37 (37)	56 (56)
Not evaluable ^c^	-	3 (3)	12 (12)
ORR	35 (35)	52 (52)	32 (32)

Data are numbers (percentage). CR: complete response; PR: partial response; SD: stable disease; PD: progressive disease; ORR overall response rate (CR+PR). ^a^ Required a second session of RT in 8 patients or surgical excision of a remaining lesion in 5 patients. ^b^ Required a second session of RT in 6 patients. ^c^ Bone lesions in radiated areas or absence of non-radiated areas or of RECIST measurable lesions outside of radiated areas.

## Data Availability

This study analyses data originating from patients’ records and completely open access to data was not included in our submission to the Ethics committee. The anonymised datasets of this study in xlsx format are therefore available from the corresponding author upon reasonable request.

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
