# Peer review of "Efficacy of Large Use of Combined Hypofractionated Radiotherapy in a Cohort of Anti-PD-1 Monotherapy-Treated Melanoma Patients"

_cancers, 2022, doi:10.3390/cancers14174069_

Round 1

Reviewer 1 Report

The manuscript of Saiag et al. is a summary of retrospective clinical analysis of a relatively large cohort of advanced melanoma patients, nearly half of which received radiotherapy, mostly after a confirmed failure of anti-PD-1 therapy. Despite limitations noted by the authors themselves and the absence of clear statistical divide between different treatments, this is a helpful contribution pointing at the value of radiotherapy in combination with checkpoint inhibitors and relatively high survival and therapeutic response levels reported therein testify to the benefit of combined treatment regimens and suggest that they can be further improved. It can be published after a minor language check to eliminate occasional word misuse (e.g., 'Could large use' on line 85 must be 'Could a widespread use' or 'both situations' on line 270 should be 'a combination of both').

Author Response

“The manuscript of Saiag et al. is a summary of retrospective clinical analysis of a relatively large cohort of advanced melanoma patients, nearly half of which received radiotherapy, mostly after a confirmed failure of anti-PD-1 therapy. Despite limitations noted by the authors themselves and the absence of a clear statistical divide between different treatments, this is a helpful contribution pointing at the value of radiotherapy in combination with checkpoint inhibitors and relatively high survival and therapeutic response levels reported therein testify to the benefit of combined treatment regimens and suggest that they can be further improved. It can be published after a minor language check to eliminate occasional word misuse (e.g., 'Could large use' on line 85 must be 'Could a widespread use' or 'both situations' on line 270 should be 'a combination of both').”

Response:

We appreciate this comment. We have checked the language with a native English-speaking colleague. We have included a clean version and a version with track changes in our revision package.

Reviewer 2 Report

Saiag et al. performed a cohort study showed that the large use of hypofractionated radiotherapy combined with anti-PD-1 induced 28 high rates of complete response.This study presented fine quality in manuscript-preparing and logic. Some points below should be still concerned: 

1) indicate the study design in the abstract;

2) please offer the information of sample power in the methods;

3) multi-variable regression in a follow-up study with time-series data need cox regression, rather than logistic regression.

Author Response

“Saiag et al. performed a cohort study that showed that the large use of hypofractionated radiotherapy combined with anti-PD-1 induced 28 high rates of complete response. This study presented fine quality in manuscript preparing and logic. Some points below should be still concerned: 

1) indicate the study design in the abstract;

Response: done. We added the term retrospectively before “cohort” in the abstract.

2) please offer the information of sample power in the methods;

Response: done. We added “The sample size calculation showed that with >200 patients treated, of whom ≥100 received concurrent radiotherapy, we had enough power to demonstrate an increase of the CR rate from 15-20% to 30%”.

3) multi-variable regression in a follow-up study with time-series data need cox regression, rather than logistic regression.

Response: We disagree with the referee's comment. A logistic regression calculates the probability of an event happening based on the factors you feed into your model, and it uses a logit transform to give you those probabilities. A Cox regression (or Cox Proportional Hazard model) is quite different. It is used to explore the relationship between the 'survival' of a subject and the explanatory variables. It operates like a linear regression except that the response variable ? is the hazard function at a given time ?.

Unlike logistic regression, this model is dependent on time, which means the hazard of an 'event' happening changes with time.

The analysis we performed looked at factors explaining why some patients achieved complete + partial response, and others did not. This is not a time-dependent variable, unlike PFS or OS. We do think that the use of logistic regression was correct for the analysis we performed.